# Conformal Time-Series Forecasting

**Kamilė Stankevičiūtė**
University of Oxford
University of Cambridge
ks830@cam.ac.uk

**Ahmed M. Alaa**
University of California, Los Angeles
ahmedmalaa@ucla.edu

**Mihaela van der Schaar**
University of Cambridge
University of California, Los Angeles
The Alan Turing Institute
mv472@cam.ac.uk

## Abstract

Current approaches for (multi-horizon) time-series forecasting using recurrent neural networks (RNNs) focus on issuing point estimates, which are insufficient for informing decision-making in critical application domains wherein uncertainty estimates are also required. Existing methods for uncertainty quantification in RNN-based time-series forecasts are limited as they may require significant alterations to the underlying architecture, may be computationally complex, may be difficult to calibrate, may incur high sample complexity, and may not provide theoretical validity guarantees for the issued uncertainty intervals. In this work, we extend the inductive conformal prediction framework to the time-series forecasting setup, and propose a lightweight uncertainty estimation procedure to address the above limitations. With minimal exchangeability assumptions, our approach provides uncertainty intervals with theoretical guarantees on frequentist coverage for *any* multi-horizon forecast predictor and *any* dataset. We demonstrate the effectiveness of the conformal forecasting framework by comparing it with existing baselines on a variety of synthetic and real-world datasets.

## 1   Introduction

Time-series forecasting tasks are central to a broad range of application domains, including stock price predictions [1, 2], service demand forecasting [3, 4], and medical prognoses [5–7]. Recurrent neural networks (RNNs) and their variants (e.g., LSTM, GRU, etc.) constitute an instrumental class of models that are most commonly used to carry out time-series forecasting tasks [8, 9]. These models, however, are usually used to issue *point* predictions—i.e., singular estimates of the future values of a time-series. In many high-stakes applications—such as finance and medicine—these are not enough; estimates of uncertainty are also required for accurate risk assessment and decision-making [10]. For example, clinical practitioners need to make treatment decisions accounting for all potential scenarios, where less likely scenarios may have graver consequences and require more care compared to the more likely scenarios [11, 12].

While various methods for uncertainty estimation in standard feed-forward neural networks have been recently proposed [13–15], equivalent methods for RNN-based time-series models are still under-explored. Existing solutions include Bayesian recurrent neural networks [16–18], quantile regression models [3, 19], latent variable models with deep state-space architectures [6, 20], and post-hoc uncertainty estimates using bootstrapping, jackknife or other ensembling procedures [21–23]. Each of these solutions has its own limitations: Bayesian models may be difficult to calibrate,

35th Conference on Neural Information Processing Systems (NeurIPS 2021).

quantile predictors may "overfit" their uncertainty estimates, and bootstrapping methods scale poorly for RNNs with large number of parameters. Almost all existing methods share at least one of the two major drawbacks: (1) they require substantial modifications to the underlying model architecture, and (2) they provide no theoretical guarantees on frequentist coverage, any of the exceptions being computationally intractable.

We aim to address the above limitations by adapting *conformal prediction* (CP) [24, 25]—a framework used to derive prediction intervals with *guaranteed* finite-sample frequentist coverage—to the time-series forecasting setup. CP has originally been designed to construct prediction intervals for *scalar* targets; on the other hand, observations and predictions in time-series forecasting involve temporally dependent, potentially multivariate *sequences* that are not, in general, directly comparable due to differences in observation lengths, irregular frequencies, non-stationarity, and other variations in temporal dynamics (comparison between training points being a key step in CP). We extend CP to a novel, computationally efficient *conformal forecasting* framework that can leverage *any* underlying point forecasting model to produce multi-step prediction intervals with coverage guarantees across the prediction horizon. We focus on RNN-based conformal forecasting architectures, which we call *conformal forecasting RNNs* (CF-RNNs), and explore their effectiveness in providing *valid* and *efficient* coverage intervals.

## 2 Related Work

Most previous work in the area of uncertainty quantification in deep learning focuses on feed-forward neural network models. Much less work has been done on uncertainty estimation for time-series models. In what follows, we discuss previous methods developed for uncertainty estimation for RNNs, which we also summarise in Table 1.

Table 1: Overview of the most popular RNN-based probabilistic forecasting methods.

| Method | Paradigm | Architecture | Time-series observations | Frequentist coverage |
|---|---|---|---|---|
| Bayesian RNNs [16–18] | Bayesian | Built-in | Multiple | — |
| Monte Carlo dropout [26] | Bayesian (approx.) | Built-in | Multiple | — |
| MQ- [3], SQF-RNN [19] | — | Built-in | Multiple | — |
| BJ-RNN [21] | Frequentist | Post-hoc | Multiple | $1 - 2\alpha$ |
| EnbPI [27] | Frequentist (approx.) | Ensemble | Single | $1 - \alpha$ |
| CF-RNN (proposed) | Frequentist | Post-hoc | Multiple | $1 - \alpha$ |

**Bayesian RNNs** [16–18] extend the ideas of Bayesian inference to RNN models, expressing the model (*epistemic*[1]) uncertainty through distributions on model parameters [28, 29]. Exact Bayesian inference quickly becomes infeasible, however; various approximations based on Markov chain Monte Carlo [30–33] or variational inference [34–37] are needed. Bayesian neural networks depend on significant changes in the underlying model architecture (at least doubling the number of parameters), and rely on a good choice of prior (which may be challenging in practice). While simplifying techniques such as Monte Carlo dropout [26] (with RNN-specific adaptation in Gal and Ghahramani [38]) exist, they are often difficult to calibrate [21].

**Quantile RNNs** can be viewed as a deep neural network extension of quantile regression [39] for sequential data: instead of returning a series of point estimates across the prediction horizon, quantile RNNs learn the prediction intervals directly, with upper and lower bounds of the forecast as separate prediction targets. The standard approach to achieve this is to use the appropriate pinball loss function as the objective. While successful applications of this approach in time-series forecasting exist [3], naively learning individual bounds may have problems such as quantile crossing; more recent approaches Gasthaus et al. [19] resolve this by fitting the entire quantile function. Quantile RNNs are additionally at risk of *quantile overfitting* due to poor sample complexity [21].

**Ensembles** are based on the principle of training and combining multiple models, e.g. deep ensembles trained on different random initialisations [40, 41], or models retrained on partial datasets (jackknife or bootstrap resampling-based RNNs, [22, 21]). Deep neural network ensembles are in general not mathematically principled for uncertainty quantification [42]; while resampling-based

---

[1]Contrast with *aleatoric* uncertainty of the data [12].

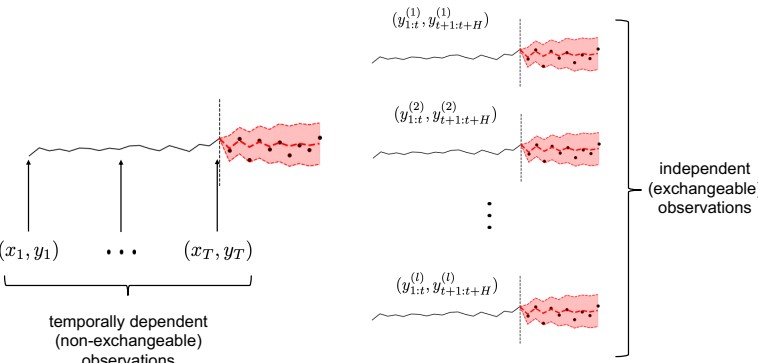

Figure 1: **Time-series observation paradigms.** (Left) The dataset is assumed to comprise a *single* time-series, with observations being individual time-steps within the time-series. These observations are temporally dependent. (Right) The dataset consists of a *set* of independent time-series, where the entire series is treated as an observation. Independence of time-series implies their exchangeability.

models resolve this and provide post-hoc frequentist coverage guarantees, they are instead limited in their time and space complexity. For example, exact inference on the state-of-the-art blockwise jackknife RNN (BJ-RNN) model [21] takes $O(P^3)$ time for $P$ parameters, and even with simplifying approximations—which in turn deteriorate accuracy—the model does not scale beyond small datasets.

**Conformal prediction (CP)**  For a given significance level (error rate) $\alpha$, the goal of CP [24, 25] is to return a *prediction region* $\Gamma^\alpha$ that is guaranteed to contain the true value with probability of at least $(1 - \alpha)$. In regression problems (such as time-series forecasting), CP is modified to work inductively using an additional *calibration set* and an *underlying model*—an approach called *inductive* conformal prediction (ICP) [43, 44]. Little work has been done applying (I)CP methods for time-series forecasting; the main challenge is that CP assumes *exchangeability*, where any permutation of the dataset observations is equiprobable. However, the time-steps within a time-series are inherently non-exchangeable due to temporal dependencies (Figure 1, left); naively applying CP to derive forecast intervals from a given time-series is therefore not methodologically valid and lacks the validity guarantees. One notable exception is the EnbPI model [27], which bypasses the exchangeability assumption (introducing some others) and uses an ensemble of bootstrapped estimators to provide *approximately* valid intervals. However, we argue that learning from a *single* time-series—while useful in cases where indeed only one time-series is available—may not be optimal in settings where datasets contain *multiple* time-series, the *shared* patterns of which could potentially be exploited (Figure 1, right). To the best of our knowledge, no existing method has applied CP to the latter forecasting setting (despite it being more methodologically grounded); yet the datasets of multiple time-series are increasingly common and useful [45].

## 3 Conformal forecasting RNNs (CF-RNNs)

In this Section, we introduce the conformal forecasting RNN (CF-RNN) model. We start off by formalizing the multi-horizon time-series forecasting problem in Section 3.1, and providing the necessary background on inductive conformal prediction (ICP) for regression tasks in Section 3.2. We introduce the details of the conformal forecasting procedure in Section 3.3.

### 3.1 Multi-horizon time-series forecasting

Let $y_{t:t'} = (y_t, y_{t+1} \ldots, y_{t'})$ be a time-series of $d$-dimensional observations $y_t, \ldots, y_{t'} \in \mathbb{R}^d$ that start at time step $t$ and end at time step $t'$. A *multi-horizon time-series forecast* predicts future values

$$\hat{y}_{(t'+1):(t'+H)} = (\hat{y}_{t'+1}, \ldots, \hat{y}_{t'+H}) \in \mathbb{R}^{H \times d}, \tag{1}$$

given the history of observed values $y_{1:t'}$, where $H$ is the number of steps to be predicted (the *prediction horizon*). For critical applications, we are interested in the *uncertainty* associated with the forecast—for each time step $h$ in the prediction horizon, we would like to obtain prediction intervals of the form $[\hat{y}_{t+h}^L, \hat{y}_{t+h}^U]$, $h \in \{1, \ldots, H\}$, so that the ground truth value $y_{t+h}$ is contained in the interval with a sufficiently high probability. We fix a desired *significance level* (or *error rate*) $\alpha$, such that the ground-truth values of the *entire time-series trajectory* are contained within the intervals; i.e.,

$$\mathbb{P}\left[y_{t+h} \in [\hat{y}_{t+h}^L, \hat{y}_{t+h}^U], \, \forall h \in \{1, \ldots, H\}\right] \geq 1 - \alpha. \tag{2}$$

## 3.2 Inductive conformal prediction (ICP)

Given a set of observations $\mathcal{D} = \{(\mathbf{x}^{(i)}, y^{(i)})\}_{i=1}^l$ and a new example $\mathbf{x}^{(l+1)}$, the ICP procedure [24, 46, 47] returns a prediction interval $\Gamma^\alpha$ such that the property of *validity* is satisfied:

**Property 1. (Validity)** *Under the exhangeability assumption, any conformal predictor will return the prediction region $\Gamma^\alpha(\mathbf{x}^{(i)})$ such that the probability of error $y^{(l+1)} \notin \Gamma^\alpha(\mathbf{x}^{(l+1)})$ is not greater than $\alpha$. Alternatively:*

$$\mathbb{P}[y^{(l+1)} \in \Gamma^\alpha(\mathbf{x}^{(l+1)}) \mid \mathcal{D}] \geq 1 - \alpha. \tag{3}$$

The conformal prediction framework is *distribution-free* (i.e. it does not have any assumptions on the distribution of the underlying data $\mathcal{D}$), and applies to *any* underlying predictive model as long as the *exchangeability* assumption is satisfied:

**Assumption 1. (Exchangeability)** *In a dataset of $l$ observations $\{(\mathbf{x}^{(i)}, y^{(i)})\}_{i=1}^l$, any of its $l!$ permutations are equiprobable. Note that independent identically distributed (iid) observations satisfy exchangeability.*

The inductive[2] variant of CP operates by splitting the training set into the *proper* training set of size $n$ and a *calibration* set of size $m$: $\mathcal{D} = \mathcal{D}_{\text{train}} \cup \mathcal{D}_{\text{cal}}$. The proper training set is used to train the *underlying* (auxiliary) model $M$, and the calibration set is used to obtain the *nonconformity scores*, which measure how unusual is the given example compared to previously observed data. While CP guarantees validity for *any* nonconformity score (including a random number generator), the most commonly used nonconformity score in regression is of the form

$$R_i = A(\mathcal{D}, (\mathbf{x}^{(i)}, y^{(i)})) = \Delta(M(\mathbf{x}^{(i)}|\mathcal{D}), y^{(i)}), \tag{4}$$

where $\Delta$ is some *distance metric*. While any choice for $M$ is *valid*, the best architecture depends on the dataset and the problem. When $\Delta(\hat{y}, y) = |\hat{y} - y|$, the nonconformity score $R_i = |\hat{y}^{(i)} - y^{(i)}|$ corresponds to the *residual error* between the prediction of the underlying model and the true label.

The resulting *empirical nonconformity score distribution* $\{R_i\}_{i=1}^l$ is used to compute a *critical nonconformity score* $\hat{\varepsilon}$, which corresponds to the $\lceil(m + 1)(1 - \alpha)\rceil$-th smallest residual [48]. For a new example $\mathbf{x}^{(l+1)}$, the prediction interval is then:

$$\Gamma^\alpha(\mathbf{x}^{(l+1)}) = [\,\hat{y}^{(l+1)} - \hat{\varepsilon}, \hat{y}^{(l+1)} + \hat{\varepsilon}\,], \tag{5}$$

with $\hat{y}^{(l+1)} = M(\mathbf{x}^{(l+1)})$.

## 3.3 CF-RNN: ICP for multi-horizon RNNs

So far we have considered the case when the labels $y \in \mathbb{R}$ are *scalar*, but multi-horizon time-series forecasts return $H$ ($d$-dimensional) values (in this work, we focus on $d = 1$; extending the results to multivariate time-series is left for future work). We extend the ICP framework to handle the

---

[2]As opposed to the standard "transductive" setting, following the categorisation in Zeni et al. [48]. Alternative definitions for "transductive" predictors exist; for further discussion see Vovk [49].

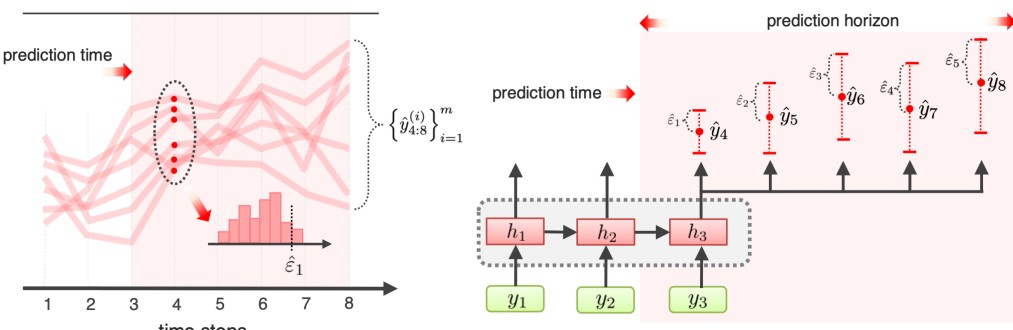

Figure 2: **CF-RNN uncertainty estimation procedure**. (a) The calibration set is used to obtain the empirical distribution of nonconformity scores $\hat{\varepsilon}_h$, and its appropriate quantile is selected depending on the desired target coverage level. (b) Critical nonconformity scores are used to obtain the prediction interval.

multi-horizon forecasting setup, while maintaining the validity of the resulting multi-horizon forecast intervals—we call this the *conformal forecasting* framework.

Let $\mathcal{D}$ be the set of exchangeable observations of the form $(y_{1:T}, y_{T+1:T+H})$, where $y_{1:T}$ is the time-series consisting of $T$ observed steps, and $y_{T+1:T+H}$ is the $H$-step forecast. Note that the label $y_{T+1:T+H}$ is now an $H$-dimensional value, in contrast with the scalar $y$ value from before. Due to the sequential nature of the task, we will use an RNN as the underlying model $M$. We set $M$ to produce multi-horizon forecasts *directly* (where at each time step $t$, all values of the $H$-step target $y_{t+1:t+H}$ are predicted at the same time from a single embedding) rather than *recursively* (where a single prediction is obtained at a time, and successive values are obtained by iteratively feeding them back into the RNN). We motivate our choice of the direct strategy by its robustness to error accumulation [50, 3], and conditionally independent predictions given the state of $M$ (which will be important for theoretical guarantees as discussed below). We now replace the single-dimensional nonconformity score defined earlier by its $H$-dimensional counterpart,

$$R_i = \left[ |y_{t+1}^{(i)} - \hat{y}_{t+1}^{(i)}|, \ldots, |y_{t+H}^{(i)} - \hat{y}_{t+H}^{(i)}| \right]^\top, \tag{6}$$

where $\left[ \hat{y}_{t+1}^{(i)}, \ldots, \hat{y}_{t+H}^{(i)} \right]^\top = M(y_{1:t}^{(i)})$. Since the $H$ conditionally independent predictions are obtained from the same embedding, we apply Bonferroni correction to the critical calibration scores in order to maintain the desired error rate $\alpha$. In particular, the original $\alpha$ is divided by $H$, so that the critical nonconformity scores $\hat{\varepsilon}_1, \ldots, \hat{\varepsilon}_H$ become the $\lceil (m+1)(1 - \alpha/H) \rceil$-th smallest residuals in the corresponding nonconformity score distributions. The resulting set of prediction intervals is therefore

$$\Gamma_1^\alpha \left( y_{(1:t)}^{(l+1)} \right), \ldots, \Gamma_H^\alpha \left( y_{(1:t)}^{(l+1)} \right), \tag{7}$$

where

$$\Gamma_h^\alpha \left( y_{(1:t)}^{(l+1)} \right) = \left[ \hat{y}_{t+h}^{(l+1)} - \hat{\varepsilon}_h, \hat{y}_{t+h}^{(l+1)} + \hat{\varepsilon}_h \right] \quad \forall h \in \{1, \ldots, H\}. \tag{8}$$

In summary, the conformal forecasting RNN (CF-RNN) model consists of an RNN issuing point forecasts, and a conformal forecasting procedure to derive the uncertainty. The entire procedure for constructing prediction intervals in CF-RNN is illustrated in Figure 2 and summarized in Algorithm 1. Finally, we show the theoretical motivations behind our approach via the following Theorem, which provides validity for intervals obtained with the conformal forecasting procedure.

**Theorem 1. (Conformal forecasting validity)**  *Let $\mathcal{D} = \left\{ \left( y_{1:t}^{(i)}, y_{t+1:t+H}^{(i)} \right) \right\}_{i=1}^l$ be the dataset of exchangeable time-series observations and their $H$-step forecasts obtained from the same underlying*

*probability distribution. Let $M$ be the recurrent neural network predicting $H$-step forecasts using the direct strategy. For any significance level $\alpha \in [0, 1]$, the intervals obtained with the ICP-based conformal forecasting algorithm will have the error rate of at most $\alpha$; alternatively,*

$$\mathbb{P}\left(\forall h \in \{1, \dots, H\}.\ y_{t+h} \in [\hat{y}_{t+h} - \hat{\varepsilon}_h, \hat{y}_{t+h} + \hat{\varepsilon}_h]\right) \geq 1 - \alpha. \tag{9}$$

The proof follows from conditional validity of ICP in Vovk [51] and Boole's inequality. The full statement and detailed proof is provided in Appendix A.

---

**Algorithm 1** Conformal forecasting RNN (CF-RNN)

---

1: **Input:** A trained model $M$ producing $H$-step forecasts,
2:          calibration dataset $\mathcal{D}_{\text{cal}} = \left\{(y_{1:t}^{(i)}, y_{t+1:t+H}^{(i)})\right\}_{i=1}^{m}$, target error rate $\alpha$.
3: **Output:** Critical nonconformity scores $\hat{\varepsilon}_1, \dots, \hat{\varepsilon}_H$.

---

4: Initialize $\varepsilon_1 = \{\}, \dots, \varepsilon_H = \{\}$.
5: **for** $i = 1$ **to** $m$ **do**
6:      $\hat{y}_{t+1:t+H}^{(i)} \leftarrow M(y_{1:t}^{(i)})$.
7:      **for** $h = 1$ **to** $H$ **do**
8:          $\varepsilon_h \leftarrow \varepsilon_h \cup \{|\hat{y}_{t+h}^{(i)} - y_{t+h}^{(i)}|\}$.
9:      **end for**
10: **end for**
11: **for** $h = 1$ **to** $H$ **do**
12:      (Bonferroni and finite sample correction)
13:      $\hat{\varepsilon}_h \leftarrow \lceil(m+1)(1 - \alpha/H)\rceil$-th smallest residual in $\varepsilon_h$.
14: **end for**
15: **return** $\hat{\varepsilon}_1, \dots, \hat{\varepsilon}_H$.

---

16: For a new time-series example $y_{1:t}^*$:
17:      $\hat{y}_{t+1:t+H}^* \leftarrow M(y_{1:t}^*)$.
18:      **return** intervals $\hat{y}_{t+1}^* \pm \hat{\varepsilon}_1, \dots \hat{y}_{t+H}^* \pm \hat{\varepsilon}_H$.

---

## 4 Experiments

In this section, we showcase the performance of the conformal forecasting RNN (CF-RNN) model against three baselines: the frequentist blockwise jackknife RNN (BJ-RNN) [21], the multi-quantile RNN (MQ-RNN) [3], and the Monte Carlo dropout-based RNN (DP-RNN) [26]. We chose these baselines as the most popular and representative examples of the different paradigms for uncertainty estimation (frequentist, quantile prediction and Bayesian uncertainty estimation for BJ-RNN, MQ-RNN and DP-RNN respectively). All architectures use LSTM as the underlying recurrent neural network, and are adapted to produce direct multi-horizon forecasts. We first present the performance of CF-RNNs on synthetic data with controlled properties. Since BJ-RNNs do not scale to larger real datasets, we use smaller synthetic datasets to provide the comparison of BJ-RNNs with the other methods. Finally, we compare the performance of CF-RNNs with the remaining two baselines on three real-world medical datasets. The code is available at github.com/kamilest/conformal-rnn.

### 4.1 Synthetic data

We first generate the synthetic time-series consisting of two components: the autoregressive process determining the trend of the time-series, and the noise process representing the inherent uncertainty of the dataset.[3] For a time-series of length $T$, this is expressed mathematically as:

---

[3]Additional experiments showcasing performance on time-series with explicit seasonal (periodic) components of different frequencies are discussed in Appendix B.2.

$$y_t = \sum_{k=0}^{t} a^k \cdot x_k + \epsilon_t, \forall k \in \{1, \ldots, T\},\tag{10}$$

where $x_t \sim \mathcal{N}(\mu_x, \sigma_x^2)$, $a = 0.9$ is the memory parameter and $\epsilon_t \sim \mathcal{N}(0, \sigma_t^2)$ is the noise process. We consider five time-dependent noise variance profiles, $\sigma_t^2 = 0.1tn$ and five static noise variance profiles $\sigma_t^2 = 0.1n$, for $n = \{1, \ldots, 5\}$.

## 4.2 Results

We train the models on 2000 training sequences (with CF-RNNs splitting this dataset into 1000 true training and 1000 calibration sequences) for the two noise variance profiles. We aim to forecast prediction intervals for $H$ future values $y_{T+1:T+H}$ for a default coverage rate of 90% ($\alpha = 0.1$). Here, $T = 15$ and $H = 5$. The RNN hyperparameters for the networks underlying the uncertainty estimation models are fixed in order to ensure fair comparison, and largely follow those provided in previous work [21]. These are detailed in the Appendix B along with the time-series model parameters. Where possible,[4] we repeat the experiments five times with a new randomly generated dataset, reporting the variation in empirical joint coverage over the different realisations.

Table 2: Comparison of joint coverages produced by CF-RNNs and competing baselines on autoregressive series with static or time-dependent noise profiles. Where possible, empirical joint coverages are aggregated over repeated trials with randomly generated datasets.

| Noise mode | | Empirical joint coverage | | | |
| --- | --- | --- | --- | --- | --- |
| | | CF-RNN | BJ-RNN | MQ-RNN | DP-RNN |
| Static $\sigma_t^2 = 0.1n$ | $n = 1$ | $92.8 \pm 0.8\%$ | 100% | $65.0 \pm 2.7\%$ | $5.4 \pm 0.5\%$ |
| | $n = 2$ | $94.0 \pm 0.4\%$ | 100% | $65.6 \pm 3.4\%$ | $5.6 \pm 1.0\%$ |
| | $n = 3$ | $94.6 \pm 1.6\%$ | 100% | $66.4 \pm 1.9\%$ | $5.0 \pm 0.9\%$ |
| | $n = 4$ | $94.3 \pm 1.4\%$ | 100% | $65.2 \pm 4.4\%$ | $4.7 \pm 1.0\%$ |
| | $n = 5$ | $94.3 \pm 1.4\%$ | 100% | $67.2 \pm 1.6\%$ | $4.2 \pm 1.0\%$ |
| Time-dependent $\sigma_t^2 = 0.1tn$ | $n = 1$ | $92.7 \pm 1.3\%$ | 99.4% | $63.4 \pm 1.5\%$ | $2.5 \pm 1.1\%$ |
| | $n = 2$ | $92.4 \pm 0.9\%$ | 100% | $60.9 \pm 1.9\%$ | $0.4 \pm 0.2\%$ |
| | $n = 3$ | $90.9 \pm 1.3\%$ | 100% | $57.2 \pm 2.1\%$ | $0.3 \pm 0.2\%$ |
| | $n = 4$ | $90.6 \pm 1.2\%$ | 97.0% | $57.1 \pm 3.7\%$ | $0.0 \pm 0.1\%$ |
| | $n = 5$ | $91.1 \pm 0.7\%$ | 99.4% | $58.6 \pm 2.1\%$ | $0.1 \pm 0.1\%$ |

Table 3: Prediction interval widths of CF-RNNs and competing baselines on synthetic datasets of autoregressive time-series with static or time-dependent noise profiles. The mean and standard deviation are reported over all prediction horizons and random seeds.

| Noise mode | | Interval widths | | | |
| --- | --- | --- | --- | --- | --- |
| | | CF-RNN | BJ-RNN | MQ-RNN | DP-RNN |
| Static $\sigma_t^2 = 0.1n$ | $n = 1$ | $16.45 \pm 3.69$ | $98.45 \pm 25.95$ | $9.47 \pm 1.99$ | $2.82 \pm 0.33$ |
| | $n = 2$ | $16.97 \pm 3.34$ | $32.53 \pm 2.92$ | $9.63 \pm 1.85$ | $2.95 \pm 0.37$ |
| | $n = 3$ | $17.12 \pm 3.50$ | $35.82 \pm 1.59$ | $9.72 \pm 1.92$ | $2.77 \pm 0.37$ |
| | $n = 4$ | $17.34 \pm 3.77$ | $33.83 \pm 2.49$ | $9.71 \pm 1.80$ | $2.87 \pm 0.35$ |
| | $n = 5$ | $16.97 \pm 3.27$ | $51.23 \pm 3.21$ | $9.84 \pm 1.99$ | $2.85 \pm 0.38$ |
| Time-dependent $\sigma_t^2 = 0.1tn$ | $n = 1$ | $19.80 \pm 3.61$ | $27.09 \pm 1.16$ | $11.50 \pm 1.66$ | $3.01 \pm 0.35$ |
| | $n = 2$ | $25.74 \pm 3.32$ | $104.85 \pm 5.68$ | $15.45 \pm 1.68$ | $3.15 \pm 0.37$ |
| | $n = 3$ | $32.70 \pm 3.97$ | $36.45 \pm 1.25$ | $20.05 \pm 2.02$ | $3.62 \pm 0.33$ |
| | $n = 4$ | $40.74 \pm 4.10$ | $33.24 \pm 2.32$ | $25.02 \pm 2.11$ | $3.91 \pm 0.45$ |
| | $n = 5$ | $49.00 \pm 5.58$ | $51.45 \pm 5.37$ | $30.55 \pm 2.54$ | $4.15 \pm 0.57$ |

Tables 2 and 3 compare the joint coverage uncertainty intervals of the models. Both CF-RNN and BJ-RNN empirically surpass the target joint coverage of 90% ($\alpha = 0.1$) in both static and time-dependent noise settings, satisfying the finite-sample frequentist coverage guarantees as required.

---

[4]BJ-RNNs were not retrained beyond a single random seed due to limited resources.

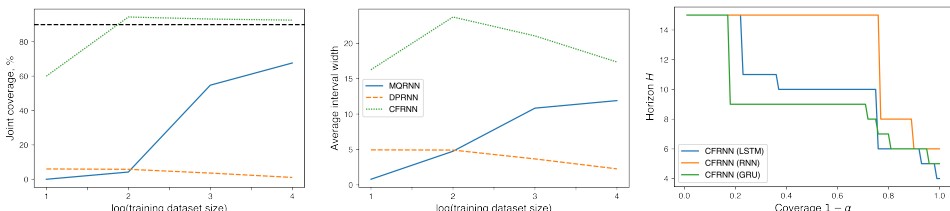

Figure 3: **Trade-offs between the dataset size, joint coverage and interval widths.** (Left and Middle) The relationship between the training dataset size, joint coverage rate (left) and average interval width (middle) for CF-RNN, MQ-RNN and DP-RNN baselines. (Right) The trade-off between the coverage rate and prediction horizon for a fixed prediction interval width in CF-RNN models with different types of the underlying RNN.

Table 3 additionally illustrates that CF-RNN intervals adapt to the properties of the temporal dynamics of the dataset: when the noise is static (and the time-series more predictable), CF-RNN prediction interval widths do not change much with increasing base variance; on the other hand, when the noise profile is time-dependent—so that inherent noise of the time-series accumulates in addition to uncertainty of the model itself—the average intervals get wider with increasing dataset uncertainty. The other frequentist baseline—BJ-RNN—has markedly wider intervals than those of CF-RNN. This might be important to maintain the perfect coverage; however, we argue that as long as the coverage rate surpasses target coverage, the intervals should be as efficient (narrow) as possible to be the most informative for decision making. (Consider that infinite intervals would have perfect coverage but would not be informative.) In addition, BJ-RNNs take prohibitively long to compute (the reason for which they contain only a single seed and will also be excluded from the comparisons on real data). Conversely, the ICP procedure only requires running the trained RNN model on a calibration set (where the model can be calibrated for any desired coverage simultaneously, with no additional computational cost), at which point adding uncertainty intervals to a prediction takes constant time. On the other hand, baselines following the alternative (non-frequentist) paradigms—MQ-RNN and DP-RNN—both fail to achieve target coverage, sometimes reporting coverage rates as low as zero. For this reason, while the two models also come with narrower (more efficient) intervals, lack of coverage guarantees makes them less useful in high stakes real-world applications.

Experiments on the data with controlled properties provide additional insight on the trade-offs between the desired coverage rate and how far into the future can the predictions be reliably made. These trade-offs are shown in Figure 3 as applied to the datasets with time-dependent noise variance profile $\sigma_t^2 = 0.1t$. The left and middle panels show the average performance of the CF-RNN, MQ-RNN and DP-RNN baselines depending on the training dataset size. CF-RNN is the only model to achieve and maintain the required joint coverage rate with finite number of examples; additionally, with more data (larger calibration datasets), the distribution of nonconformity scores can be specified more accurately, so the width of the intervals decreases. Finally, in the panel on the right we fix the prediction interval width and for each horizon $H$ compute the largest coverage level $1 - \alpha$ maintained by CF-RNN. As shown in the Figure, low target coverage levels allow us to make valid predictions far into the future, and ideal coverage levels can only be achieved with horizons near the prediction point. The overall trend is maintained for every recurrent neural network model $M$.

## 4.3   Experiments on real data

We now demonstrate the effectiveness of our procedure on real-world time-series. We train the proposed CF-RNN architecture as well as the MQ-RNN and DP-RNN baselines on three datasets summarised in Table 4. For the first task, we use the data from the Medical Information Mart for Intensive Care (MIMIC-III) [52] dataset, where we forecast daily observations of white blood cell counts of varying lengths. For the second task, we use the electroencephalography (EEG) dataset from the UCI machine learning repository [53], where we forecast trajectories of downsampled EEG signals obtained from healthy subjects exposed to three types of visual stimuli. For the final task, we forecast daily COVID-19 cases within the United Kingdom local authority districts. All datasets are publicly available and the medical data is anonymised. We selected these datasets to represent a variety of scenarios of real time-series: the numbers of available training instances span different orders of magnitude (from hundreds to tens of thousands), the datasets have varying observation sequence lengths, different stationarity properties (e.g. the COVID-19 time-series are synchronous—each time step representing the same point in time—the others are not), different

noise profiles (e.g. EEG signal data will inherently have higher frequencies than MIMIC-III white blood cell count data), and different target prediction horizons. We note that COVID-19 dataset is especially challenging as the forecasts contain a wave of infections and lockdowns, with the wave starting at different points for every region. Details on datasets and their preprocessing are provided in Appendix C.

Table 4: Dataset properties. The number in parentheses under the training example column indicates how many examples were used for training when a calibration set was required, such as in the case of CF-RNNs.

| Dataset | # Training sequences | Window length $T$ | Prediction horizon $H$ |
|---|---|---|---|
| MIMIC-III [52] | 3823 (2000) | [3, 47] | 2 |
| EEG [53] | 19200 (15360) | 40 | 10 |
| COVID-19 [54] | 300 (200) | 100 | 50 |

Table 5: Uncertainty forecasting model performance on three real-world datasets. Coverage refers to joint coverage (higher is better), and is averaged over the random splits of the dataset and training seeds. Prediction interval lengths (lower is better) are averaged over the prediction horizons and random seeds.

| Model | MIMIC-III | | EEG | | COVID-19 | |
|---|---|---|---|---|---|---|
| | Coverage | CI/PI lengths | Coverage | CI/PI lengths | Coverage | CI/PI lengths |
| DP-RNN | $40.2 \pm 13.9\%$ | $3.59 \pm 0.90$ | $3.3 \pm 0.7\%$ | $7.39 \pm 0.74$ | $0.0 \pm 0.0\%$ | $61.18 \pm 32.37$ |
| MQ-RNN | $89.3 \pm 1.2\%$ | $16.16 \pm 3.92$ | $48.0 \pm 4.0\%$ | $21.39 \pm 2.36$ | $15.0 \pm 5.9\%$ | $136.56 \pm 63.32$ |
| CF-RNN | $94.0 \pm 1.2\%$ | $20.59 \pm 3.10$ | $96.5 \pm 1.0\%$ | $61.86 \pm 18.02$ | $89.7 \pm 5.3\%$ | $733.95 \pm 582.52$ |

Performance of the models is summarised in Table 5. We note that the underlying LSTM model of the proposed CF-RNN architecture had the same hyperparameters as the competing baselines, yet fewer training instances (as some of the examples are used for the calibration procedure). Despite this, CF-RNN obtains the highest coverage for all datasets, and is the only model to empirically achieve the target joint coverage rates. While this seems to disproportionately affect the efficiency of CF-RNN intervals (as these are the widest), the predictions are indeed reliable across the range of datasets and scenarios. On the other hand, the baseline models seem to have competitive coverage with better efficiency in some settings (e.g. MIMIC-III), yet revert to unreliable predictions in less certain scenarios (e.g. COVID-19). In other words, CF-RNN adapts its prediction interval widths to reliably match the required target coverage, increasing the width for unpredictable datasets.

Finally, we briefly explore the importance of and motivation behind the Bonferroni correction of the error rate in the CF-RNN calibration procedure. Table 6 shows that calibration scores without Bonferroni correction generally lead to poor *joint* coverage, even though *independent* coverage rates normally achieve the target coverage (most notable exception being the COVID-19 dataset selected for its forecasting difficulty).

Table 6: Bonferroni-corrected and uncorrected empirical coverages of the CF-RNN model. Joint coverage is aggregated over the different random seeds; independent coverages present the range of observed values across all horizons and all random seeds.

| Model | MIMIC-III | | EEG | | COVID-19 | |
|---|---|---|---|---|---|---|
| | Joint | Independent | Joint | Independent | Joint | Independent |
| CF-RNN | $94.0 \pm 1.2\%$ | [93.8%, 96.8%] | $96.5 \pm 1.0\%$ | [98.3%, 99.8%] | $89.7 \pm 5.3\%$ | [87.5%, 100.0%] |
| Uncorrected | $89.0 \pm 1.4\%$ | [89.0%, 91.4%] | $59.4 \pm 2.4\%$ | [85.5%, 91.6%] | $55.5 \pm 8.0\%$ | [77.5%, 98.8%] |

## 5  Conclusion

In this paper, we extended the ICP framework to the multi-horizon time-series forecasting problem, providing a lightweight algorithm with theoretical guarantees for frequentist coverage. Extending from the initial investigation presented in Appendix D, future work would focus on increasing the overall efficiency of prediction intervals by reducing their width, which would involve making them more adaptive to individual observations.

**Acknowledgments**

The authors would like to thank the reviewers for their helpful comments. This work was supported by AstraZeneca, the US Office of Naval Research (ONR) and the National Science Foundation (NSF, grant number 1722516).

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
