# A Proof of Theorem 1

**Theorem 1.** *Let $\mathcal{D} = \{(y_{1:t}^{(i)}, y_{t+1:t+H}^{(i)})\}_{i=1}^{l}$ be the dataset of exchangeable time-series observations and their $H$-step forecasts obtained from the same underlying probability distribution. Let $M$ be the recurrent neural network predicting $H$-step forecasts using the direct strategy. For a target coverage level $\alpha \in (0, 1)$, the intervals obtained by the ICP-based conformal forecasting algorithm satisfy*

$$\mathbb{P}\left(\forall h \in \{1, \ldots, H\}.\ y_{t+h} \in [\hat{y}_{t+h} - \hat{\varepsilon}_h, \hat{y}_{t+h} + \hat{\varepsilon}_h]\right) \geq \alpha$$

**Proof.** Due to the direct forecasting strategy, every step in the horizon can be treated as a separate inductive conformal predictor that uses the same underlying model $M$ (with the final predictions derived from the internal state being independent) and the same dataset $\mathcal{D}$. The independent validity of each of the $H$ ICPs follows from Proposition 1 in Vovk [51]. Setting the error rate of each of the $H$ ICPs to $(1 - \alpha)/H$ and applying Boole's inequality we obtain that the combined error rate of the $H$-step forecaster is $1 - \alpha$, as required. □

# B Synthetic data experiments

Table 7: Hyperparameters for the synthetic data experiments.

| Parameter | Value |
|---|---|
| Training samples | 1000 |
| Calibration samples | 1000 |
| Test samples | 500 |
| Sequence length $T$ | 15 |
| Prediction horizon $H$ | 5 |
| Autoregressive mean $\mu_x$ | 1 |
| Autoregressive variance $\sigma_x^2$ | 2 |
| Periodicity $s$ | None |
| Amplitude $u$ | 5 |
| Epochs | 1000 |
| Batch size | 100 |
| Embedding size | 20 |
| Learning rate | 0.01 |
| Underlying RNN type | LSTM |
| Target coverage $1 - \alpha$ | 90% |

## B.1 Qualitative results for static and time-dependent noise settings

See Figure 4.

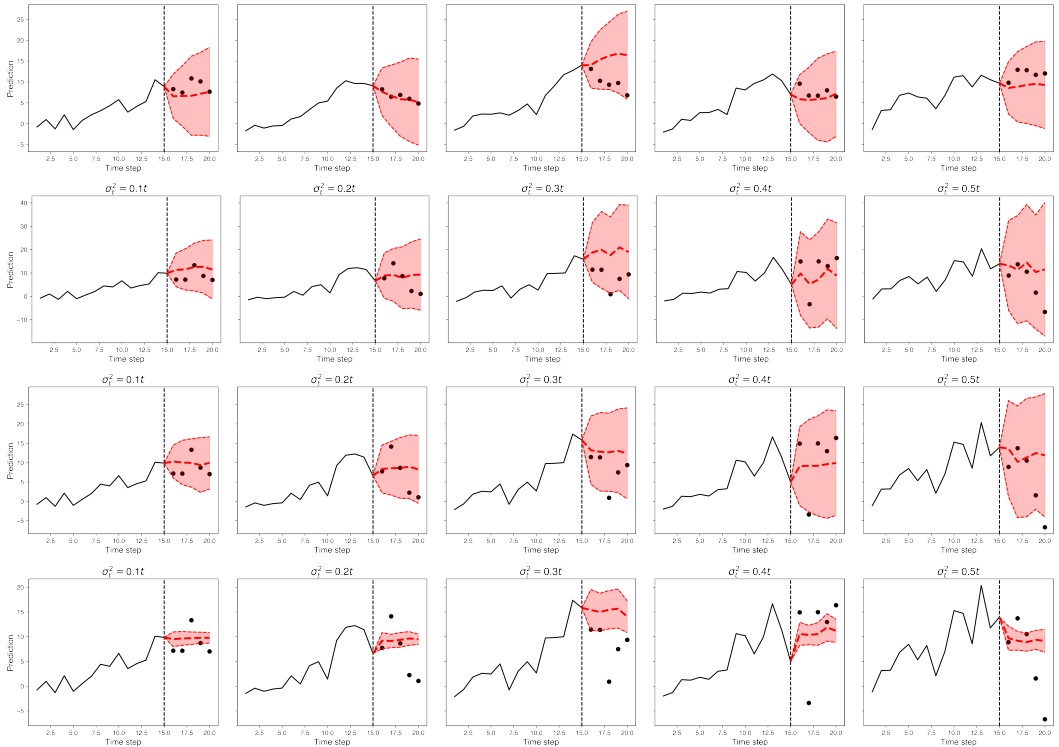

Figure 4: **Example prediction intervals for the different uncertainty baselines.** From top to bottom: 1) CF-RNN for the static noise variance setting; 2) CF-RNN for the time-dependent noise variance setting; 3) MQ-RNN for the time-dependent noise variance setting; 4) DP-RNN for the time-dependent noise variance setting.

## B.2 Seasonal and varying-length synthetic datasets

We now generate the synthetic time-series consisting of three components: the autoregressive process determining the trend of the time-series, the seasonal process representing periodic fluctuations in the time-series values, and the noise process representing the inherent uncertainty of the dataset, expressed mathematically as:

$$y_t = \sum_{k=0}^{t} a^k \cdot x_k + \gamma_t + \epsilon_t, \forall k \in \{1, \ldots, T\} \tag{11}$$

where $x_t \sim \mathcal{N}(\mu_x, \sigma_x^2)$, $a = 0.9$ is the memory parameter, $\gamma_t$ is the stochastic seasonal process representing periodic fluctuations and $\epsilon_t \sim \mathcal{N}(0, \sigma_t^2)$ is the noise process.

The periodic component is defined following the quasi-random walk model in Durbin and Koopman [55] (Equations (3.7) and (3.8)): we define $\gamma_t = \sum_{j=1}^{\lfloor s/2 \rfloor} \gamma_{jt}$ where $\gamma_{j,t+1} = \gamma_{jt} \cos \lambda_j + \gamma_{jt}^* \sin \lambda_j + w_{jt}$ and $\gamma_{j,t+1}^* = -\gamma_{jt} \sin \lambda_j + \gamma_{jt}^* \cos \lambda_j + w_{jt}^*$, $s$ is the period length, $\lambda_j = 2\pi/s$ and $w_{jt}, w_{jt}^* \sim \mathcal{N}(0, u)$ for some amplitude $u$.[5]

To stress-test the conformal prediction model, we challenged it with two datasets of two different frequencies (2 and 10, for the mean observation length of 20), that 1) consisted of variable-length time-series, 2) had high noise amplitude (5 compared to 1 in the other synthetic datasets), 3) had each example start at a random phase of the periodic component.

Table 8: Empirical joint coverage of CF-RNN for the datasets with asynchronous, out-of-phase examples with dynamic series lengths, averaged across prediction horizons and reported as mean $\pm$ std over five random seeds.

| | Empirical joint coverage | | | |
|---|---|---|---|---|
| Periodicity | CF-RNN | Adaptive CFRNN | MQ-RNN | DP-RNN |
| 2 | 75.9 ± 38.0% | 75.9 ± 38.0% | 81.2 ± 1.2% | 9.4 ± 7.8% |
| 10 | 76.3 ± 38.1% | 76.3 ± 38.2% | 68.8 ± 2.0% | 2.8 ± 5.7% |

Table 9: Mean interval widths (reported as mean $\pm$ std over the prediction horizon) as predicted by the CF-RNN model. The results are reported for five random seeds; empty spaces denote seeds where the training procedure for the given model was unstable.

| | Interval widths | | | |
|---|---|---|---|---|
| Periodicity | CF-RNN | Adaptive CF-RNN | MQ-RNN | DP-RNN |
| 2 | 80.61 ± 13.34 | 81.54 ± 13.34 | 105.36 ± 5.34 | 3.35 ± 0.37 |
| 10 | — | — | 104.69 ± 4.45 | 34.48 ± 1.26 |
| 2 | 90.16 ± 12.06 | 89.83 ± 12.01 | 106.17 ± 4.50 | 23.88 ± 0.31 |
| 10 | 136.09 ± 17.27 | 136.17 ± 18.18 | 105.20 ± 3.66 | 3.49 ± 0.19 |
| 2 | 94.09 ± 13.45 | 94.89 ± 13.69 | 109.02 ± 3.99 | 21.02 ± 0.27 |
| 10 | 177.77 ± 8.90 | 179.17 ± 9.06 | 106.15 ± 4.16 | 3.43 ± 0.67 |
| 2 | 160.06 ± 5.22 | 160.87 ± 5.66 | 106.57 ± 3.64 | 25.06 ± 0.27 |
| 10 | 176.95 ± 8.44 | 179.38 ± 10.17 | 104.68 ± 4.82 | 3.11 ± 0.16 |
| 2 | — | — | 106.08 ± 3.64 | 3.43 ± 0.30 |
| 10 | 118.98 ± 11.60 | 118.62 ± 11.92 | 104.73 ± 3.37 | 3.36 ± 0.43 |

We additionally present qualitative results on periodic datasets with different frequencies. Despite the larger errors in the higher-frequency series in Figure 5, conformal prediction intervals cover the ground truth in both cases. Prediction interval widths also increase moving further away from the prediction step and with increasing dataset variance, reflecting increasing uncertainty.

---

[5]For implementation details, see also https://www.statsmodels.org/devel/examples/notebooks/generated/statespace_seasonal.html

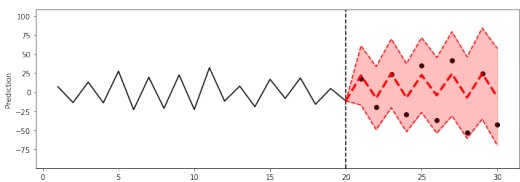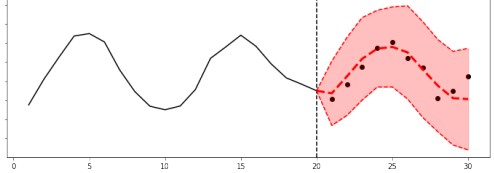

Figure 5: Example prediction intervals for the CF-RNN model and datasets with different periodic components.

## C Real-world datasets

**MIMIC-III** We collect the data of patients on antibiotics from the MIMIC-III dataset [52], and filter out the sequences of total length at least 5, resulting in 4323 sequences. From these sequences we pick out the white blood cell (high) count as the feature for the univariate time-series. We split the sequences randomly into training, calibration and test datasets. We pick the constant time horizon of 2, which is to account for the shortest sequences being of length 5, and use the rest of the sequence as model input.

**EEG** The EEG dataset, available at https://archive.ics.uci.edu/ml/datasets/ EEG+Database, was used as the source for the EEG signal time-series. The dataset contains the data for control and alcoholic subjects responding to a visual stimulus of three types. We used the medium version of the dataset, involving 10 control and 10 alcoholic subjects, though for the experiments we only used the control subjects—from the summaries provided, control subject EEG responses seemed to be more difficult to predict. Each subject had repeated trials for every type of stimulus, and each trial had a time-series for the 64 channels obtained from their corresponding sensor. We treated every individual trial and each of the 64 channels as a separate time-series example, resulting in 19200 sequences in the training set. To keep training efficient, we downsampled the sequences (normally of length 255) to a total length 50, which we further split into the training sequence of 40 and prediction horizon 10. The training and test dataset splits are readily provided in the UCI repository, and for repeated trials we used different subsets for calibration and different model training seeds.

**COVID-19** The data is available at https://coronavirus.data.gov.uk/. We picked the data of different regions of the same country in order to follow the exchangeability assumption as closely as possible, while the data from different countries risks having much larger distribution shifts due to a large variation of factors like government lockdown policies. Given the setup of the conformal prediction framework, we looked for the data that would have a sufficiently large number of independent sequences—the lower tier local authority split gives a total number of 380 sequences, which over repeated trials we would randomly split into the test set of 80 sequences and the the rest between the training and calibration sets. We picked the data of daily new cases over the course of 150 days starting mid-September 2020 and ending mid-February 2021, which we further split into the input sequence of 100 examples (ending Christmas 2020) and using the remaining 50 days as the testing sequence. We chose these dates to capture interesting properties of changing government lockdown policies and so that the two waves are separated between the observed and the to-be-predicted sequence.

**Hyperparameters** The training hyperparameters (Table 10) mostly follow those provided in previous work [21] and are kept the same for new experiments in order to ensure fair comparison between the baselines. For the CoRNN model, the total training data available was split between the training and calibration sets, and the other baselines used all available training data to train the underlying RNN model.

Table 10: Training hyperparameters for the real-world datasets.

| Parameter | MIMIC-III | EEG | COVID-19 |
|---|---|---|---|
| Training samples | 3823 (2000) | 19200 (15360) | 300 (200) |
| Calibration samples | 1823 | 3840 | 100 |
| Test samples | 500 | 19200 | 80 |
| Sequence length $L$ | [3, 47] | 40 | 100 |
| Prediction horizon $S$ | 2 | 10 | 50 |
| Epochs | | 1000 | |
| Batch size | | 150 | |
| Embedding size | | 20 | |
| Learning rate | | 0.01 | |
| Dropout probability (for DP-RNN) | | 0.5 | |
| Underlying RNN type | | LSTM | |
| Target coverage $1 - \alpha$ | | 90% | |

## D   Adaptive CF-RNNs

In standard conformal prediction, once the underlying model is trained on the training dataset and calibrated on the calibration datasetm, the prediction interval width, $2\hat{\varepsilon}$, will be the same for *every* subsequent example (in case of CF-RNN, the intervals will be horizon-specific, but not vary across examples). While this gives *valid* intervals, they are not as *efficient*, as the widths are determined by the residuals of the most difficult examples with the largest residuals in the dataset. *Normalisation* [46], then, tries to return the interval widths that are *example-specific*; i.e. if the model knows that the example is "simple" to forecast, the intervals will be more narrow; if the example is very unusual, the intervals will be wider.

This is achieved through a modification to the nonconformity score $R_i$ as follows:

$$R_i = \frac{|y^{(i)} - M(\mathbf{x}^{(i)})|}{\exp(\mu^{(i)}) + \beta}, \tag{12}$$

where the numerator is as before, and the denominator captures the "difficulty" of the example through an estimate of the residual error:

$$\mu^{(i)} = \log |y^{(i)} - M(\mathbf{x}^{(i)})|, \tag{13}$$

and $\beta$ is the sensitivity parameter. The estimates $\mu^{(i)}$ are obtained through training another model (often a neural network such as a multilayer perceptron) on the examples in the proper training set and their log residuals: $\left\{ (\mathbf{x}^{(i)}, \log |y^{(i)} - \hat{y}^{(i)}|) \right\}_{i=1}^{n}$. We learn the logarithms of errors for them to both have a smaller range across examples, and to enforce the errors to be positive once they are raised to the exponent as $R_i$ is computed. As the difficulty score $\mu^{(i)}$ is example-specific, for the new example $\mathbf{x}^{(l+1)}$ the new interval obtained is

$$\Gamma^{\alpha}(\mathbf{x}^{(l+1)}) = [\hat{y}^{(l+1)} - \hat{\varepsilon}(\exp(\mu^{(l+1)}) + \beta), \hat{y}^{(l+1)} + \hat{\varepsilon}(\exp(\mu^{(l+1)}) + \beta)]. \tag{14}$$

This is analogously extended to the forecast horizon-specific set of $\hat{\varepsilon}_h$ in the conformal time-series forecasting procedure.

We carry out the experiments on the synthetic datasets discussed in the main paper, to investigate the effects of the normalised nonconformity scores on the quality of prediction intervals. Since CF-RNN is designed to work on time-series of different lengths, the normalisation network is also trained on a recurrent neural network, contrary to the recommendation in literature (see e.g. Papadopoulos and Haralambous [46]) to use simple predictors. We use the same parameters for the normalisation RNN as the underlying model $M$; we set the sensitivity parameter $\beta = 1$.

The results of these experiments are shown in Tables 11 and 12. We observe that, with the simplistic hyperparameter setting, we achieve the *opposite* effect from the intended one: while the intervals

Table 11: **Empirical joint coverage of CF-RNN and Adaptive CF-RNN baselines run on autoregressive synthetic datasets** (averaged across prediction horizons); reported as mean ± std over five random seeds (excluding unstable seeds).

| Noise mode | | Empirical joint coverage | |
|---|---|---|---|
| | | CF-RNN | Adaptive CF-RNN |
| Static $\sigma_t^2 = 0.1n$ | $n=1$ | $92.8 \pm 0.8\%$ | $93.6 \pm 0.4\%$ |
| | $n=2$ | $94.0 \pm 0.4\%$ | $94.7 \pm 1.0\%$ |
| | $n=3$ | $94.6 \pm 1.6\%$ | $93.6 \pm 0.3\%$ |
| | $n=4$ | $94.3 \pm 1.4\%$ | $94.2 \pm 1.3\%$ |
| | $n=5$ | $94.3 \pm 1.4\%$ | $93.7 \pm 1.1\%$ |
| Time-dependent $\sigma_t^2 = 0.1tn$ | $n=1$ | $92.7 \pm 1.3\%$ | $93.3 \pm 1.2\%$ |
| | $n=2$ | $92.4 \pm 0.9\%$ | $92.2 \pm 1.3\%$ |
| | $n=3$ | $90.9 \pm 1.3\%$ | $91.3 \pm 1.1\%$ |
| | $n=4$ | $90.6 \pm 1.2\%$ | $91.4 \pm 1.0\%$ |
| | $n=5$ | $91.1 \pm 0.7\%$ | $89.8 \pm 1.4\%$ |

Table 12: **Mean interval widths of CF-RNN and Adaptive CF-RNN baselines run on autoregressive synthetic datasets** (reported as mean ± std over the prediction horizon and the five random seeds, excluding those seeds where results were unstable).

| Noise mode | | Interval widths | |
|---|---|---|---|
| | | CF-RNN | Adaptive CF-RNN |
| Static $\sigma_t^2 = 0.1n$ | $n=1$ | $16.45 \pm 3.69$ | $21.05 \pm 4.96$ |
| | $n=2$ | $16.97 \pm 3.34$ | $21.90 \pm 5.68$ |
| | $n=3$ | $17.12 \pm 3.50$ | $20.81 \pm 4.45$ |
| | $n=4$ | $17.34 \pm 3.77$ | $23.29 \pm 5.12$ |
| | $n=5$ | $16.97 \pm 3.27$ | $21.56 \pm 5.68$ |
| Time-dependent $\sigma_t^2 = 0.1tn$ | $n=1$ | $19.80 \pm 3.61$ | $24.95 \pm 5.28$ |
| | $n=2$ | $25.74 \pm 3.32$ | $30.75 \pm 4.46$ |
| | $n=3$ | $32.70 \pm 3.97$ | $40.16 \pm 6.68$ |
| | $n=4$ | $40.74 \pm 4.10$ | $48.23 \pm 6.32$ |
| | $n=5$ | $49.00 \pm 5.58$ | $53.92 \pm 6.71$ |

stay valid, they become *less*, rather than *more*, efficient. One reason for this is that the underlying and normalisation RNNs are both noisy, which makes the residuals difficult to learn. This results in noisy normalisation estimates that do not help with reducing the interval widths. Another reason, as discussed in Romano et al. [56], is that the standard normalisation procedure is not adaptive to *heteroscedastic* datasets, where the variance of the data depends on the time-step, which is the case in these synthetic datasets. Making the interval widths more robust to the noisy underlying and normalising RNNs as well as more adaptive to heteroscedasticity is an interesting problem for future work.