# OpenReview forum: "Conformal Time-series Forecasting"
_NeurIPS.cc/2021/Conference — NeurIPS 2021 Poster_

### Official Review · Reviewer_vdnC · 2021-07-16

**Rating:** 6
**Confidence:** 3

**Summary:**

This paper uses conformal learning for inductive time series forecasting that can produce accurate uncertainties. The method have theoretical guarantee but is post-hoc.




**Limitations And Societal Impact:**

Yes

**Main Review:**

How is the computation efficiency of this post-hoc method? How can it be generalized? This method needs an pre-determined p-value and should run multiple times if multiple p-values are used? This needs multiple times of the computation?

A missing reference is [1] which presents that the conformal time series forecaster produce similar results compared with the baseline in terms of the pinball loss. This NeurIPS submission does not compare with other uq/calibration methods and please give some clarification on whether this NeurIPS submission share the same conclusion with [1]?

It can only be used in inductive setting. I am wondering whether  can use other units, such as CNN or transformer?

Does this method require an extra calibration/conformal dataset?

[1] http://oa.upm.es/57817/1/TFM_JAKUB_KOWALCZEWSKI.pdf

----POST REBUTTAL---

Author rebuttal have resolved most of my questions and therefore I increase my score.

**Time Spent Reviewing:**

3

---

> ### Author Response · Authors · 2021-08-10
> **Response to Comments by Reviewer vdnC**
>
> Thank you for your comments. We provide a point-by-point response below.
>
> ## Post-hoc model
> We would like to note that we consider the model being post-hoc one of the **benefits** of the framework, as this is exactly what makes it easily applicable to any underlying model that supports multi-horizon forecasting (e.g. RNN, Seq2Seq models, etc.) Unlike in other models, where changing the prediction objective (e.g. in quantile forecasters) or the underlying architecture (e.g. the Bayesian frameworks) might change the accuracy of the model and/or make it intractable, our model can be applied to any underlying multi-horizon forecaster depending on the practitioner’s needs, without undermining the accuracy of the point estimates while bringing out the theoretical guarantees.
>
> ## Computational efficiency
>
> The main bottleneck of our proposed conformal forecasting framework is the complexity of training the underlying model, such as the RNN in our experiments. In order to compute the critical calibration scores, we only need to run the RNN on the examples in the calibration set, the complexity of which is $\sim O(mT)$ where $m$ is the number of examples in the calibration set and $T$ is the length of the longest time-series example. Therefore the calibration procedure, as well as the overall CoRNN, model is efficient and even beats the other baselines, i.e. the BJ-RNN baseline that was intractable for most of our experimental setups. We also note that for multiple confidence levels we **do not need to retrain** the model; having obtained the empirical distribution of the calibration scores (see Figure 2a of the manuscript and Line 14 of Algorithm 1), instead of returning a single quantile $\hat{\varepsilon}_h$, we can return several, corresponding to different coverage levels.
>
> ## Missing reference
>
> Thank you for pointing out the missing reference to Kowalczewski. Please note that our problem setup is different from the one proposed in the paper, where an observation is considered to be a single time-step within a single time-series and where the conformal prediction framework is not actually applicable without breaking the theoretical guarantees, its main benefit (as pointed out in Kowalczewski, Sections 3.4.1 and 3.4.2). Instead, we use a dataset of many time-series (e.g. many independent patients) and treat the *entire* time-series as a single observation, which, in addition to satisfying the assumptions of the conformal prediction framework and therefore providing guarantees on performance, allows us to learn from the information contained in the other time-series (e.g. patterns shared between patients). We **further explain the distinction in the following figure**: https://i.imgur.com/s05sWAb.png.
>
> ## Calibration/conformal dataset
>
> This is correct, our model splits the training set into a proper training set and a calibration set, as we explain in Lines 155-156 of the manuscript and Line 2 of Algorithm 1.

---

> > ### Comment · Reviewer_vdnC · 2021-08-19
> > **missing reference issue**
> >
> > Thanks for your detailed reply which resolved most of my questions.
> >
> > For the missing reference issue, from your reply and figure, I can see the difference of the two settings but still do not know why the entire time series observation could bring benefits.

---

> > > ### Author Response · Authors · 2021-08-22
> > > **Response to Reviewer vdnC: benefits of using datasets comprising multiple time-series**
> > >
> > > Thank you for your question. In the response below, we detail why observation of the entire time-series is not only *beneficial* for producing better forecasts, but also *necessary* for the validity of the resulting uncertainty intervals, and how this is achieved through a dataset of multiple time-series.
> > >
> > > We again compare our work with the reference to Kowalczewski. In Kowalczewski, the *single* time-series (which is the *entire* dataset) is split into two contiguous sections, one for training and one for calibration; **see figure here:** https://i.imgur.com/uj2kNca.png. We argue that such treatment of a single time-series is **not valid**, because we **cannot meaningfully permute the time-steps** of the series; we also *cannot* benefit from observing an *incomplete* time-series when we want to produce good forecasts of future values of that series, especially if we have not observed the *most recent* values before the forecasting point. In contrast, our work uses as much information as it possibly can before producing the forecast; **see figure here:** https://i.imgur.com/u6co721.png. In our treatment of the data, **we can produce better forecasts (because we learn from multiple time-series and observe them entirely) while also having valid uncertainty intervals (as the time-series within the training/calibration/test datasets can be observed in any order).**
> > >
> > > We would also like to reiterate that **the setting when multiple time-series are available is becoming increasingly important and useful.** In addition to the healthcare domain explored in our work—where each patient in the hospital has their own time-series—Salinas et al. (https://arxiv.org/abs/1704.04110) list additional examples, such as energy consumption forecasting in individual households, or server load forecasting for independent servers in a data center. In these applications, **individual time-series are inter-related**, and instead of fitting a *separate* model with its *unique* parameters for *each* patient/household/server, we can **learn from the patterns that are *shared* between the examples**. This is especially useful when some time-series may contain less information than others; for example, if we want to forecast patient health from observing just a few time-steps, we can do that better if we leverage information from related patients that we have observed for longer. Another illustrative example is that, by fitting a unique model to each patient in isolation, it would be impossible to learn how to forecast things like death of the patient (because the patient can only die once and only in the part of the time-series that the model is supposed to forecast). On the other hand, if we have trained the model on the *full* time-series of some patients, the model will have seen death in the *training* dataset, and would be able to forecast that for people in the *test* dataset.
> > >
> > > We hope that this is helpful and are happy to answer any further questions.

---

> > > ### Author Response · Authors · 2021-08-26
> > > **Dear Reviewer vdnC**
> > >
> > > Dear Reviewer vdnC,
> > >
> > > We hope our response to additional questions has been helpful and were wondering if there is anything else we could do to further improve the paper. :)

---

> > ### Author Response · Authors · 2021-08-30
> > **Dear Reviewer vdnC**
> >
> > Dear Reviewer vdnC
> >
> > We would like to thank you again for your review and follow up to ask if there is anything else we could do to improve the paper and its score and if there are any more remaining questions or comments. :)
> >
> > Thank you!

---

### Official Review · Reviewer_Z5gg · 2021-07-16

**Rating:** 6
**Confidence:** 4

**Summary:**

This work extend the idea of conformal prediction to time series setting successfully.

**Main Review:**

Originality: This work is the first practically applicable work that conformal prediction has been applied beyond the typical case of scalar target Y .

Quality: The work is technically solid under the time series wise exchangability assumption authors made. The methodology is simple enough, basically apply conformal prediction individually/separately on each prediction time point.
- However, I suspect that two claims in the beginning (in line 38-46) have been attained or not as they promised. For example, how this method handle different time length, frequency and asynchronosness that existing methods fails under regression setting? It would be great the claims are clearly demonstrated.
- What if comparing with quantile forecaster? Spline Quantile Function by Gasthaus et al http://proceedings.mlr.press/v89/gasthaus19a/gasthaus19a.pdf does not suffer the overfitting (quantile crossing) the authors worry about.

Clarity/Questions:
- This methodology seems not to be limited to RNN predictor. Any seq2seq forecaster should be applicable. Correct me if I am wrong.
- One missing reference is 'Exact and Robust Conformal Inference Methods for Predictive Machine Learning With Dependent Data' https://arxiv.org/abs/1802.06300 which covers time series data
- In many time series data exchangability often heavily breaks. For example, hierarchical dataset . In that case, direct quantile regression may work but this method may fail. In addition, in this case, Chen et al [26] and Chernozhukov et al (reference above) could be more relevant.


Significance: Even with several questions mentioned above to be resolved, this work is theoretically sound under the assumption and potential to be impactful to the practitioner, providing codes.



**Time Spent Reviewing:**

2.5

---

> ### Author Response · Authors · 2021-08-10
> **Response to Comments by Reviewer Z5gg**
>
> Thank you so much for your review. We provide a point-by-point response to your comments below.
>
> ## Different time lengths, frequencies, asynchronousness
>
> This point is in fact closely related to your other comment on the independence of the conformal forecasting framework from the underlying point forecast predictors (e.g. RNN, Seq2Seq, etc). While in the paper we present a proof-of-concept demonstration and use an RNN as a popular and natural choice of predictor, conformal forecasting can be used with any Seq2Seq model that is able to process time-series data and provide multi-horizon forecasts. RNNs can handle different time lengths by design, as well as datasets containing time-series of different frequencies and starting at different phases of the periodic component (asynchronousness). At the same time, it is possible to replace the RNN with a different model that could handle unaligned, non-stationary datasets in a more sophisticated fashion, e.g. models that explicitly handle observations in continuous time, e.g. a Phased LSTM model. In the final version of the paper, we will make sure to make these points clearer.
>
> To clearly demonstrate the applicability of our procedure to these settings and to stress-test the models for possible limitations, we **include an additional experimental setup** (**see Tables R3 and R4 in the combined official response**), where we add random phase to the sinusoidal time-series to capture asynchronousness.
>
>
> ## Comparison with spline quantile forecasters
>
> Thank you for this observation, we will add a note on the architectures that are able to solve the quantile crossing problem in Lines 75-76 of the final paper. Please note that, despite these improvements, quantile forecasters still do not provide theoretical guarantees—which is the main contribution of our proposed framework—and can still suffer from overfitting in the sense of generalisation performance. In particular, the accuracy of quantiles depends on the sample size, with prediction intervals being unreliable in small sample datasets, where uncertainty quantification matters the most. Low sample efficiency of quantile forecasters is further discussed in e.g. ref. [21] of the main paper.
>
> ## Applicability to any seq2seq forecaster
>
> Yes, our procedure is applicable to any Seq2Seq forecaster. We will clarify this in the final manuscript.
>
> ## Exchangeability assumption
>
> In our problem setup, we assume that distinct time-series are statistically independent (as we **illustrate in the figure** here: https://i.imgur.com/s05sWAb.png). For instance, in a clinical application, different time-series will correspond to different patients whose data will be independent. To the best of our knowledge, this assumption will naturally hold for most practical applications. We are unclear as to what you mean by a hierarchical dataset—we can explain whether our procedure can generalize to these datasets if you kindly clarify this question during the discussion period.
>
> ​​Thank you for pointing out the missing reference to Chernozhukov et al. This will be added to the final version of the paper.

---

> ### Author Response · Authors · 2021-08-26
> **Dear Reviewer Z5gg**
>
> Dear Reviewer Z5gg,
>
> Thank you so much again for your thoughtful review, and we hope that our answers so far have been helpful. We are following up to ask if there are any further questions or comments, or anything else we could do to further improve the paper. :)

---

> ### Author Response · Authors · 2021-08-30
> **Dear Reviewer Z5gg**
>
> Dear Reviewer Z5gg,
>
> We would like to thank you again for your review and follow up to ask if there is anything else we could do to improve the paper and its score and if there are any more remaining questions or comments. :)
>
> Thank you!

---

### Official Review · Reviewer_qsqN · 2021-07-17

**Rating:** 6
**Confidence:** 3

**Summary:**

The paper extends the conformal prediction framework, hitherto limited to models trained under the IID assumption, to the context of multi-step time series forecasting. It shows that under this framework, one can obtain reasonably well calibrated predictive intervals of the future trajectory of a time series, an important problem in decision theory.  Experiments with synthetic data and several healthcare datasets confirm the validity of the approach against baselines.

**Limitations And Societal Impact:**

The manuscript only cursorily discuss the limitations of the proposed approach. It does not address the societal impacts of the work.

**Main Review:**

The paper is in general well written and easy to follow. The main result appears novel and sound, and builds on a little-known method in ML that would benefit from greater visibility.

Some specific questions appear in the detailed comments below.

The theoretical results appear ok, although I did not assess the proofs in depth.

Even though experimental results can be strengthened (see comments below), they confirm the validity of the approach.

Assuming that questions are satisfactorily answered, the paper would make a valuable contribution to the literature. I would recommend acceptance for publication.

Detailed comments (by line or section):
- 75: Even though naïve quantile regression suffers from quantile crossing, it is straightforward to prevent it in deep learning based architectures through reparametrization of the output layer.
- Eq 1: if the observations are multidimensional with dimensionality $d$, then they will be in $R^{H \times d}$, no?
- Eq 7 is a bit hard to interpret: which examples in this calculation come from the calibation set, and which come from the training set?
- Would Eq 8 need to be modified for non-symmetrical noise models (e.g. regression with a Box-Cox transformation of the dependent variable)? Could the critical nonconformity scores be computed separately for both tails of the residuals distribution?
- Figure 2b, the illustration of the RNN makes it appear that the forecast over the prediction horizon is iterated instead of direct (as stated in the paper).
- Section 4.2: why only include BJ-RNN as a baseline model for synthetic data, and not also MQ-RNN and DP-RNN? This would give a sense of their performance profile across the board.
- 223: it feels awkward that CoRNN could have access to twice as much data as BJ-RNN. Can you arrange the experiments such that both have access to the same dataset size but use it differently to compute the coverage?
- 242: coverage should be calibrated, not “higher”: if aiming for a 90% coverage, it is a disaster to get 100% (as shown in the width of the intervals for BJ-RNN). I suggest rephrasing to avoid suggesting that higher coverage can be “better” in some interpretations.
- 248: why say CoRNN “is competitive” while it Pareto-dominates the baseline?
- Section 4.3: are you also targeting $\alpha = 0.9$ ?

**Time Spent Reviewing:**

3

---

> ### Author Response · Authors · 2021-08-10
> **Response to Comments by Reviewer qsqN**
>
> Thank you for your detailed review!
>
> Below, we provide a point-by-point response to your bulleted comments. We also highlight the changes that we will incorporate into the final manuscript in order to implement your suggestions.
>
> ## Quantile crossing in naïve quantile regression
>
> Thank you for pointing this out. We will add a note on this in Line 75 of the final paper. Please note that while quantile crossing can be avoided via more sophisticated quantile predictors, they still do not have frequentist coverage guarantees, which is the main advantage of our proposed method. Moreover, the accuracy (i.e., generalization performance) of quantile intervals will depend on the sample size, with less reliable prediction intervals in small-sample datasets—these are the data sets for which uncertainty quantification matters the most. We will highlight this in the final manuscript.
>
> ## Multidimensional observations in Eq (1)
>
> Yes, that is true—the observations will be in $\mathbb{R}^{d \times H}$ in this case.
>
> ## Interpretation of Eq (7)
>
> Thank you for this comment—Equations 6 and 7 in fact contain typos and should be modified as follows:
>
> * (line 158) $\mathbf{x}^{l+1}$
> * Eq 6: $R_i = |y^{(n+i)} - M(\mathbf{x}^{(l+1)})|,\quad  \forall i \in {1, \dots, m}$
> * (line 159) $p_{(\mathbf{x}_{l+1}, y)}$
> * Eq 7: ${p}\_{(\mathbf{x}\_{l+1}, y)} := \frac{|\{i = n+1, \dots, n+m, l+1 : R_i \geq R_{l+1}\}|}{m + 1}$
>
>
> We hope this clears some of the confusion. Otherwise, the main observation in  Equation (7) is that the training dataset is of size $l = m + n$ (line 128). It would be used for training in its entirety for the baselines, and is split into $n$ proper training examples $(x^{1}, \dots, x^{n})$ and $m$ calibration examples $(x^{n+1}, \dots, x^{m+1})$ for the conformal prediction framework. The $(l+1)$-th example is a *new* example (from the testing set); it could also be denoted as the $(n+m+1)$-th example but $(l+1)$ is a bit shorter. We will make sure to make this clear in the final paper.
>
> ## Non-symmetrical noise models
>
> Please note that our method is **distribution-free**: it does not assume any particular noise distribution and still maintains frequentist coverage guarantees. These guarantees will hold for any noise distribution, including a non-symmetric one. That being said, for a non-symmetrical aleatoric noise distribution, the *optimal* confidence intervals (in the sense of having the narrowest interval length) for an *optimal* RNN (i.e. one that perfectly predicts the expected forecast) would be non-symmetric as well. However, such an optimal RNN would not be achievable in practice as a model trained on non-symmetrical noise will have its prediction biased towards the heavier noise tail. We believe that investigating procedures for computing non-symmetric critical scores is an interesting direction for future extensions of our model, but the gains from such procedure, in terms of reduced interval width, are likely to be marginal. We will add a discussion on this issue in the final version of the paper.
>
> ## Figure 2b clarity
>
> Thank you for this observation. The **figure has been modified** as illustrated at the following link and we hope it is clearer: https://i.imgur.com/g2LnBxt.png
>
> ## MQ-RNN and DP-RNN baselines in Section 4.1
>
> Thank you for this suggestion. For completeness, **we added the MQ-RNN and DP-RNN baselines for our synthetic experiments**, and ran all these for five random seeds (while having a more consistent dataset generation process). Please find these additional results in **Tables R1 and R2 in the combined official comment above**.
>
> ## Fixing the sample size for CoRNN and BJ-RNN
>
> We have **re-run the experiments** after accounting for this suggestion; **see Tables R1 and R2 in the combined official comment above**. CoRNN now uses the same amount of data as the other baselines, deriving the calibration set from the training data rather than using a separate calibration set (note that this was already the case for the real-world datasets, however). We find that this does not decrease performance for synthetic datasets, and CoRNN still achieves the highest joint coverage.
>
> ## Comments on phrasing in Lines 242 & 248
>
> Thank you for these suggestions. We will adjust the phrasing in the final paper accordingly.
>
> ## Section 4.3 experimental setup
>
> Yes, we are targeting $\alpha=0.9$ in this Section.

---

> ### Author Response · Authors · 2021-08-26
> **Dear Reviewer qsqN**
>
> Dear Reviewer qsqN,
>
> Thank you so much again for your detailed review, and we hope that our answers so far have been helpful. We were wondering if there was anything else we could do to address any further questions or comments. :)

---

> ### Author Response · Authors · 2021-08-30
> **Dear Reviewer qsqN**
>
> Dear Reviewer qsqN,
>
> We would like to thank you again for your review and follow up to ask if there is anything else we could do to improve the paper and its score and if there are any more remaining questions or comments. :)
>
> Thank you!

---

### Official Review · Reviewer_uEcc · 2021-07-18

**Rating:** 6
**Confidence:** 4

**Summary:**

This paper proposes a conformal prediction method (CoRNN) for uncertainty estimation in time series forecasting. CoRNN is a recurrent network-based multi-step forecaster, it can also achieve a valid finite sample coverage guarantee. Compared with other baselines, CoRNN is computationally efficient, does not require any modifications to the predictive model, and provides theoretical coverage guarantees. Empirical results on simulated and real-world datasets demonstrate the superiority of CoRNN in constructing confidence intervals for multi-step forecasts.

**Limitations And Societal Impact:**

No. The authors could briefly discuss potential problems encountered when deploying the proposed method in real applications.

**Main Review:**

This paper addresses an important problem and proposes a reasonable solution. The main content is clear and well-written. All figures are nicely designed to facilitate understanding. Besides the above advantages, I have the following comments that might be worthwhile for a further discussion:

1. There are many recent works in the conformal prediction literature, such as [1, 2] that can utilize local information to achieve adaptive interval length. CoRNN uses the basic split conformal prediction [3], which can only achieve fixed interval length (i.e., $2\cdot \hat{\varepsilon}_h$) in principle. The authors could consider borrowing ideas from more recent works to build better solutions (stronger coverage with shorter interval).
2. From my perspective, it is natural for an RNN model to use the recursive method for multi-step prediction. Despite the error accumulation, interim information can be used for future forecasts. However, I'm curious how to directly obtain forecasts in all steps at once, given the RNN can only be trained in a particular horizon.
3. The authors bypass the exchangeability assumption in split conformal prediction by treating one time series as an independent data point (instead of a timestamp). But it seems like avoiding the main challenge (temporal dependency) of conformal time series forecasting problems, as discussed in EnbPI [4] and [5]. While this is acceptable for certain applications, the authors also need to specify how to deal with variable-length data.

Minors:
1. Although this is intuitive, I'm not sure the confidence interval length at each step is strictly monotonic w.r.t. forecasting horizon.
2. In Eq (7), I wonder if the denominator should be $m+1$?
3. It is better to waive the model dependency of this method, i.e., combine with any black-box forecasting models.
4. Line 76, [6] has addressed the quantile crossing problem in training quantile RNN models for time series forecasting, the authors can briefly discuss this in related works.

References:
- [1] Romano et al, Conformalized Quantile Regression
- [2] Guan et al, Conformal prediction with localization
- [3] Lei et al, Distribution-Free Predictive Inference For Regression
- [4] Xu et al, Conformal prediction interval for dynamic time-series
- [5] Chernozhukov et al, Exact and Robust Conformal Inference Methods for Predictive Machine Learning With Dependent Data
- [6] Han et al, Simultaneously Reconciled Quantile Forecasting of Hierarchically Related Time Series
- [7] Gasthaus et al, Probabilistic Forecasting with Spline Quantile Function RNNs

-----
=====Update after Rebuttal=====


I carefully went over the author's response; most of my concerns have been addressed. I am willing to increase my score. If this paper is made public, please be sure to include:
- discussion on related works [e.g., 6, 7] that addressed quantile crossing problem for time series forecasting, and how this method is better than them
- discussion on the difference between observing time stamps v.s. time-series, as well as the exchangeability assumption

Good luck!

**Time Spent Reviewing:**

3 hours

---

> ### Author Response · Authors · 2021-08-10
> **Response to Comments by Reviewer uEcc**
>
> Thank you very much for your detailed and insightful comments! Below, we provide a point-by-point response to your comments and highlight the changes that will be applied to the final manuscript in order to incorporate your suggestions.
>
> ## Adaptive interval length/normalisation
> We have **implemented an additional normalised CoRNN baseline** where the “difficulty” of examples is learned from the log residuals as discussed in ref. [30] of the main paper. Unlike in previous literature, our forecasting framework needs to handle time-series of different lengths. For this reason we cannot use simple models as normalisation networks, and instead train another RNN as an auxiliary network (with epochs and learning rates the same as the main CoRNN network, and sensitivity parameter $\beta = 1$). The results of these experiments are attached in **Tables R1 and R2 of the combined official comment** (see the CoRNN-n baseline); we observe that, with this simplistic hyperparameter setting, the variance of joint coverage increases while the interval lengths are similar. One possible reason for this might be that the underlying RNN model is noisy and unstable as the residuals for forecasting models are difficult to learn; if the “difficulty” estimates for test examples are higher than those of the critical calibration example (or if estimates are high variance), the original $\varepsilon$ could often be multiplied by a constant greater than 1, resulting in larger interval widths.
>
> ## Direct vs. recursive forecasting strategies
>
> Direct strategy returns values for multiple steps ahead while the recursive strategy returns one step ahead. It is possible to use the forecasts returned by the direct strategy in a recursive fashion, i.e. throwing away the rest of the horizon (only using the value corresponding to a single-step forecast), feeding it into the RNN, and only taking the first value again would recover the standard recursive strategy. Another way could be to feed in the entire predicted horizon sequentially (i.e. running the RNN for $H$ steps) and using all $H$ predictions would correspond to recursive prediction but in “batches” of size $H$.
>
> We have **amended our Figure 2b in the paper** in response to another review, where we demonstrate the direct forecasting strategy more precisely: https://i.imgur.com/g2LnBxt.png
>
> While the horizon is fixed (in this case $H=3$), it is still possible to take just the point estimate $\hat{y}_4$ and feed it recursively into the underlying RNN, e.g. like this: https://i.imgur.com/m9aEc1d.png
>
> ## Exchangeability; observing time-steps vs time-series
>
> This is a very good point. Please note that our handling of each single time series as an independent data point is **not a simplifying assumption**; it is the structure of the problem and the aim for a “distribution-free” and “assumption-free” procedure that works for *any* stochastic process that dictates this treatment of the data. **Not assuming anything on the relation between time-steps makes our procedure applicable to more classes of time-series problems, not the reverse**. In what follows, we explain why this is the case; we also illustrate this in the figure here: https://i.imgur.com/s05sWAb.png.
>
> In the case where an observation is a single time-step within a single time-series, conformal inference operating on a time-step-level  is not applicable, since the exchangeability assumption is indeed broken by temporal dependencies—we cannot meaningfully permute the time-steps within a time-series and randomly split them into training/calibration datasets. **Permuting time-steps will not only break the theoretical guarantees, which are the main benefit of the conformal prediction framework, but is also not conceptually sensible since the chronological order of observations is what constitutes a time-series** (though some work has been done to partially mitigate this limitation in references provided, which we will discuss in more detail in the final manuscript).
>
> In order to allow permutation of time-steps within the same series, extra assumptions on the stochastic process $X(t)$ will be needed. In particular, we will have to assume that $X(t)$ entails a strong mixing condition that makes distant time-steps uncorrelated (Similar to *Assumption 1* in *Section 4.1* in ref. [4] by Xu et al.) **Making strong mixing assumptions on the time-series data does not only limit the set of problems to which the conformal procedure would be applicable, but also puts a limit on the accuracy of long-term forecasts since it statistically decouples distant time-steps**.
>
> In addition, treating a single time-step as an observation would assume that this one time-series corresponds to the *entire dataset*. This might be true in some cases, but in scenarios where we have a dataset of many time-series (e.g. many independent patients), this is a limitation. Instead, treating the *entire* time-series as a single observation allows us to learn from the information contained in the other time-series (e.g. patterns shared between patients). In contrast, applying the former scenario in a clinical application where an observation is a single time-step within a single time-series would correspond to learning from a single patient at a time, using a different CP model for every patient (which gives us even less information to learn from), while (in naive settings) also breaking CP assumptions and guarantees.
>
> ## Varying time-series lengths
>
> One benefit of using the RNN as the underlying model is that it naturally handles time-series of different lengths by projecting them to a fixed-length embedding. Once the entire sequence is processed by the RNN (which would take a varying number of iterations depending on the length of the time-series), a $H$-step forecast is returned. If we need varying-length forecasts, we can combine this idea with the ideas from recursive forecasting above; for online forecasting we could return $H$-step predictions at every time-step. With that said, we have **added an additional experimental setup** with 1) variable-length 2) noisy 3) asynchronous time-series of different frequencies that stress-tests the conformal prediction model; **see Tables R3 and R4 in the combined official comment**. While this setup has proved to be too difficult for **all** baselines, **CoRNN still performs competitively** even without having achieved the asymptotic guaranteed coverage rates due to the low number of examples.
>
> ## Minor comments
>
> ### Non-monotonic confidence interval length
>
> In theory, confidence interval lengths need not be monotonically increasing indeed. We could think of situations where values in the horizon could be guessed with certainty. For example, if we had a dataset of patient body temperature observations where every patient eventually stabilised to a normal body temperature of 37 degrees, and if we were to issue predictions for very long horizons, the values later in the horizon would be more and more likely to be close to 37 degrees, critical $\varepsilon$’s for those steps would get smaller, and so confidence interval lengths would get smaller. However, these situations are not very common—for the above example, we would not be very interested in the asymptotic patient body temperature, and at the same time we would have a lot of uncertainty for short-term vital sign predictions. Most real-world datasets are inherently unpredictable so uncertainty would naturally build up, like it does in the synthetic data experiments we demonstrate in the paper.
>
> ### Eq (7) denominator
>
> That is correct, the denominator should be $m+1$. Thank you very much for spotting this. :)
>
> ### Waiving the model dependencies
>
> Indeed, conformal prediction is a general framework that could work for *any* predictive model. In this paper, we adapt conformal prediction to the forecasting setting so that it can handle multi-horizon outputs, but beyond that the framework retains model independence. For the proof-of-concept demonstration of the novel conformal forecasting framework, we choose RNN as the underlying model because it naturally handles sequential inputs and produces multiple outputs at each step of the sequence; however, future work could use the framework with any model that produces multi-horizon point forecasts.
>
> ### Line 76, quantile crossing problem
>
> Thank you for bringing our attention to a model solving quantile crossing, we will add this to the final version of the paper. However, we note that while quantile crossing can be avoided via more sophisticated quantile predictors, they still do not have frequentist coverage guarantees, which is the main advantage of our proposed method. Moreover, the accuracy (i.e., generalization performance) of quantile intervals will depend on the sample size, with less reliable prediction intervals in small-sample datasets—these are the data sets for which uncertainty quantification matters the most. We will bring these points up more explicitly in the final manuscript.

---

> > ### Comment · Reviewer_uEcc · 2021-08-30
> > **Reviews Updated**
> >
> > Dear Authors,
> >
> > Thank you for your detailed response. I have updated my review according to your comments. To facilitate understanding, if this paper is made public, please be sure to include:
> > - discussion on related works [e.g., 6, 7] that addressed quantile crossing problem for time series forecasting, and how this method is better than them
> > - discussion on the difference between observing time stamps v.s. time-series, as well as the exchangeability assumption
> >
> > in the main paper. Good luck!

---

> ### Author Response · Authors · 2021-08-26
> **Dear Reviewer uEcc**
>
> Dear Reviewer uEcc,
>
> Thank you so much again for your thoughtful review, and we hope that our answers so far have been helpful. We were wondering if there was anything else we could do to address any further questions or comments. :)

---

### Author Response · Authors · 2021-08-10
**Combined official response**

We thank all Reviewers for their useful comments. The following summarises and highlights the main changes we intend to incorporate in the final manuscript.

## Improving clarity

* **Discussion on quantile forecasters**. We thank Reviewers for bringing our attention to additional models for quantile forecasting. In the final manuscript, we will clarify the benefits of the conformal forecasting framework over the quantile forecasters beyond the quantile crossing problem, as highlighted in the individual responses.

* **Exchangeability assumption vs temporal dependencies**. Please note that our work focuses on scenarios where we have a dataset of time-series samples (training data), and we are interested in making forecasts for new unseen time-series samples (test data). This is a very common setup in various applications, especially in the healthcare domain where we learn about future patients from time-series observations of past patients. **This is illustrated in the figure available here:** https://i.imgur.com/s05sWAb.png

  In our setup, exchangeability applies across different independent time-series (independent patients in the example above). We cannot assume exchangeability across the time-steps within the same time-series because this goes against the main goal of forecasting. In other words, if the time-steps within a time-series are exchangeable, then it's impossible to forecast future values or learn trends from such series.

  We note that previous works have mainly focused on extending the conformal prediction framework to a setup where we have a *single* time-series as the observation, and different *time-steps* within the series represent separate observations. In this case, strong assumptions on temporal correlations need to be made, and the desired distribution-free theoretical guarantees are broken due to temporal dependencies. On the other hand, our proposed method does not entail any assumptions on temporal correlations and can learn from multiple time-series at once, which makes it more useful in many real-world settings.


* **Model dependencies**. While we use an RNN as the underlying model in this proof-of-concept demonstration, we emphasise that conformal prediction can indeed be in general applied to *any* model that supports multi-horizon forecasting.

## Additional experiments

In response to suggestions by Reviewers uEcc, qsqN, Z5gg, we have run additional experiments. The summary of changes and experiments is as follows:

* **Fair dataset sizes for CoRNN vs other baselines.** CoRNN now uses the same amount of data as the other baselines in synthetic experiments, deriving the calibration set from the training data rather than using a separate calibration set (note that the set-up was already fair for the real-world datasets). We find that this does not decrease performance for synthetic datasets, and CoRNN still achieves the highest joint coverage.
* **Normalised CoRNN baseline.** We introduce the *normalised* CoRNN baseline where the “difficulty” of examples is learned from log residuals as discussed in e.g. ref [30] of the manuscript. We train an RNN as the underlying neural network due to possibly varying time-series lengths, with epochs and learning rate the same as in the main network, and the sensitivity parameter $\beta=1$. We observe that, with this (simplistic) hyperparameter setting (given limited time for more rigorous tuning), the variance of coverage increases, with similar or larger coverage interval lengths. We believe this might be because of the noisy underlying RNN, with the residuals being difficult to learn—this results in noisy normalisation estimates that do not help with reducing the interval widths. We would aim towards improving the stability and performance of the auxiliary RNN through the discussion period or in future work.
* **MQ-RNN and DP-RNN baselines.** For completeness, we add the MQ-RNN and DP-RNN baselines for our synthetic experiments, and run all these for five random seeds, while having an improved, more consistent and reproducible dataset generation process.
* **Experiments on variable-length, asynchronous time-series.** To stress-test the conformal prediction model, we challenged it with two datasets of two different frequencies (2 and 10, for the mean observation length of 20), that 1) consisted of variable-length time-series, 2) had high noise amplitude (5 compared to 1 in the other synthetic datasets), 3) had each example start at a random phase of the periodic component. Given limited time, we did not perform any hyperparameter tuning. While this setup has proved to be too difficult for **all** baselines (which indeed is useful for highlighting how models perform in failure cases), **CoRNN is still competitive** even without having achieved the asymptotic guaranteed coverage rates due to the low number of examples.

### Table R1: Empirical joint coverage for additional baselines

Averaged across prediction horizons; reported as mean±std over five random seeds. CoRNN-n denotes the normalised CoRNN model.

| Noise mode      | CoRNN        | CoRNN-n      | MQ-RNN       | DP-RNN      |
| --------------- | ------------ | ------------ | ------------ | ----------- |
| static $n=1$    | 93.0 ± 1.6\% | 93.1 ± 1.4\% | 62.8 ± 2.2\% | 5.3 ± 1.4\% |
| static $n=2$    | 94.3 ± 0.7\% | 94.0 ± 1.5\% | 61.9 ± 2.8\% | 4.8 ± 1.2\% |
| static $n=3$    | 93.7 ± 1.8\% | 93.4 ± 1.9\% | 65.8 ± 0.4\% | 5.3 ± 1.2\% |
| static $n=4$    | 94.2 ± 1.0\% | 93.9 ± 1.2\% | 65.5 ± 1.2\% | 5.5 ± 1.6\% |
| static $n=5$    | 93.3 ± 1.8\% | 92.9 ± 1.6\% | 63.0 ± 4.7\% | 5.3 ± 1.7\% |
|                 |
| time-dep. $n=1$ | 92.9 ± 0.7\% | 92.9 ± 1.6\% | 61.7 ± 3.9\% | 3.6 ± 0.5\% |
| time-dep. $n=2$ | 91.5 ± 1.1\% | 92.0 ± 1.8\% | 58.0 ± 2.1\% | 1.5 ± 0.5\% |
| time-dep. $n=3$ | 90.8 ± 2.2\% | 90.9 ± 1.4\% | 59.4 ± 3.1\% | 0.4 ± 0.2\% |
| time-dep. $n=4$ | 91.1 ± 1.8\% | 91.4 ± 1.9\% | 57.9 ± 1.7\% | 0.3 ± 0.2\% |
| time-dep. $n=5$ | 89.4 ± 0.9\% | 90.6 ± 0.5\% | 57.2 ± 1.5\% | 0.2 ± 0.3\% |


### Table R2: Mean interval widths for additional baselines

Reported as mean±std over the prediction horizon. Results are reported for one of five available random seeds. CoRNN-n denotes the normalised CoRNN model.

| Noise mode      | CoRNN        | CoRNN-n       | MQ-RNN       | DP-RNN      |
| --------------- | ------------ | ------------- | ------------ | ----------- |
| static $n=1$    | 17.42 ± 4.40 | 20.30 ± 6.04  | 9.36 ± 1.83  | 3.10 ± 0.17 |
| static $n=2$    | 18.01 ± 4.10 | 20.55 ± 5.92  | 9.44 ± 1.81  | 2.94 ± 0.29 |
| static $n=3$    | 17.71 ± 4.35 | 19.12 ± 5.62  | 9.65 ± 1.84  | 2.94 ± 0.20 |
| static $n=4$    | 18.90 ± 5.21 | 20.89 ± 5.78  | 9.75 ± 1.75  | 3.13 ± 0.18 |
| static $n=5$    | 18.98 ± 4.37 | 19.99 ± 5.18  | 9.69 ± 1.57  | 3.16 ± 0.21 |
|                 |
| time-dep. $n=1$ | 19.08 ± 4.06 | 21.84 ± 5.23  | 10.57 ± 1.70 | 3.11 ± 0.20 |
| time-dep. $n=2$ | 22.58 ± 4.45 | 24.73 ± 5.35  | 12.54 ± 1.91 | 2.88 ± 0.23 |
| time-dep. $n=3$ | 27.15 ± 4.58 | 29.83 ± 5.97  | 15.71 ± 2.03 | 3.46 ± 0.16 |
| time-dep. $n=4$ | 32.50 ± 5.63 | 33.94 ± 6.71  | 19.45 ± 2.34 | 3.80 ± 0.18 |
| time-dep. $n=5$ | 39.17 ± 7.72 | 44.14 ± 10.98 | 22.82 ± 3.03 | 3.96 ± 0.16 |


### Table R3: Empirical joint coverage for asynchronous, varying-length examples

Empirical joint coverage for the datasets with asynchronous, out-of-phase time-series with dynamic series lengths, averaged across prediction horizons and reported as mean±std over five random seeds. CoRNN-n denotes the normalised CoRNN model.

| Periodicity | CoRNN         | CoRNN-n       | MQ-RNN       | DP-RNN      |
| ----------- | ------------- | ------------- | ------------ | ----------- |
| 2           | 75.9 ± 38.0\% | 75.9 ± 38.0\% | 81.2 ± 1.2\% | 9.4 ± 7.8\% |
| 10          | 76.3 ± 38.1\% | 76.3 ± 38.2\% | 68.8 ± 2.0\% | 2.8 ± 5.7\% |


### Table R4: Mean interval widths for asynchronous, varying-length examples

Mean interval widths (reported as mean±std over the prediction horizon). The results are reported for five random seeds; empty spaces denote seeds where the training procedure for the given model was unstable.


| Seed | Periodicity | CoRNN          | CoRNN-n        | MQ-RNN        | DP-RNN       |
| ---- | ----------- | -------------- | -------------- | ------------- | ------------ |
| 0    | 2           | 80.61 ± 13.34  | 81.54 ± 13.34  | 105.36 ± 5.34 | 3.35 ± 0.37  |
|      | 10          | ---            | ---            | 104.69 ± 4.45 | 34.48 ± 1.26 |
|      |
| 1    | 2           | 90.16 ± 12.06  | 89.83 ± 12.01  | 106.17 ± 4.50 | 23.88 ± 0.31 |
|      | 10          | 136.09 ± 17.27 | 136.17 ± 18.18 | 105.20 ± 3.66 | 3.49 ± 0.19  |
|      |
| 2    | 2           | 94.09 ± 13.45  | 94.89 ± 13.69  | 109.02 ± 3.99 | 21.02 ± 0.27 |
|      | 10          | 177.77 ± 8.90  | 179.17 ± 9.06  | 106.15 ± 4.16 | 3.43 ± 0.67  |
|      |
| 3    | 2           | 160.06 ± 5.22  | 160.87 ± 5.66  | 106.57 ± 3.64 | 25.06 ± 0.27 |
|      | 10          | 176.95 ± 8.44  | 179.38 ± 10.17 | 104.68 ± 4.82 | 3.11 ± 0.16  |
|      |
| 4    | 2           | ---            | ---            | 106.08 ± 3.64 | 3.43 ± 0.30  |
|      | 10          | 118.98 ± 11.60 | 118.62 ± 11.92 | 104.73 ± 3.37 | 3.36 ± 0.43  |

---

### Author Response · Authors · 2021-08-17
**Comment for Reviewers**

Dear Reviewers,

Thank you so much again for your insightful reviews and suggestions. We hope that our answers so far have been helpful, and would like to ask if you have any further questions to continue the discussion. :)

Thank you!

---

### Decision · Program_Chairs · 2021-09-27

**Decision:**

Accept (Poster)

**Comment:**

The paper extends the conformal prediction methodology to the time series setting, with exchangeability assumed between time series. In doing so, the proposed approach can provide uncertainty estimates with coverage guarantees.

The reviewers agree that the paper tackles a relevant problem, proposes a reasonable solution, is well written and theoretically sound. While the reviewers pointed out some areas for improvements (additional related work, improvements to the empirical evaluation), the authors were able to alleviate all major concerns during the discussion period, making this a solid contribution with the potential to spark interesting follow-up work.